# Decoupled Search for the Masses:
## A Novel Task Transformation for Classical Planning

**Primary Keywords:** *None*

## Abstract

Automated problem reformulation is a common technique in classical planning to identify and exploit problem structures. Decoupled search is an approach that automatically decomposes planning tasks based on their causal structure, often significantly reducing the search effort. However, its broad applicability is limited by the need for specialized algorithms. In this paper, we present an approach that embodies decoupled search for non-optimal planning through a novel task transformation. Specifically, given a task and a decomposition, we create a transformed task such that the state space of the transformed task is isomorphic to that of decoupled search on the original task. This eliminates the need for specialized algorithms and allows the use of various planning technology in the decoupled-search framework. Empirical evaluation shows that our method is empirically competitive with specialized decoupled algorithms and favorable to other related problem reformulation techniques.

## Introduction

Classical planning is concerned with finding a sequence of actions that transforms the initial state of a problem into a desired one. To solve planning tasks, a representation is required that allows to search for a solution within the induced state space. Both theory and practice show that the way these problems are represented has a significant impact on the performance and success rate of planning approaches.

In domain-specific settings, problem reformulations can be approached in a very targeted way. Common examples include solving puzzles such as the Rubik's Cube, where the search is not over atomic actions but over macro actions (Korf 1997). Similarly, in the design of algorithms for matrix multiplication, the search is often not in the space of arithmetic instructions, but encapsulated as a tensor decomposition (Fawzi et al. 2022; Speck et al. 2023).

It is well-known that classical planning is PSPACE-complete in general (Bylander 1994). Nevertheless, the representation and modeling of a problem can significantly affect practical performance due to aspects like accidental complexity (Haslum 2007). Thus, problem reformulation is very relevant also in domain-independent planning with the underlying idea of exploiting the inherent structure of the planning problem. Reformulation can yield an alternative state space that may differ significantly in size

and structure from the original one, while at the same time facilitating search. An example of this is the merge-and-shrink task reformulation, which was designed to search in factored transition systems (Torralba and Sievers 2019). In this work we show that decoupled state-space search, which is based on an alternative state representation, similar to binary decision diagrams (Bryant 1986; Torralba et al. 2017), can also be interpreted as a task reformulation. Decoupled search automatically decomposes a planning problem into conditionally independent leaf components with a synchronizing center factor that interacts with the leaves, allowing the search to exploit this causal relationship (Gnad and Hoffmann 2018). A major drawback of techniques like the merge-and-shrink reformulation and decoupled search is that they require specialized algorithms and implementations, because most methods are tailored to established planning formalisms. This often leads to challenges in transferring knowledge and novel techniques to these approaches.

In this paper, we show that it is possible to simulate decoupled search for non-optimal planning via a task transformation within the widely supported finite-domain representation formalism (FDR) (Helmert 2009). This alleviates the need for specialized algorithms and enables the full toolbox of planning technology in the decoupled-search framework. More precisely, we demonstrate that given a SAS$^+$ planning task (a subset of FDR) (Bäckström and Nebel 1995), we can decompose the task as usual for decoupled search and create a FDR planning task for which the induced state space is isomorphic to that of decoupled search on the original task. Thus, a search algorithm on the transformed planning task will behave in the same way as its native decoupled search counterpart. We further show that a task reformulation by Miura and Fukunaga (2017) is closely related to our work and can be placed in the framework of decoupled search. We show that our approach generalizes it in several dimensions.

Our experiments with different planning techniques demonstrate that, just as specialized decoupled-search algorithms, our task transformation performs favorably to search on the original SAS$^+$ representation and to other reformulations techniques in multiple domains. It is even competitive with a native implementation on a large number of domains. This highlights the versatility and usefulness of our approach, which embodies the idea and workings of decoupled search through a task transformation.

# Background

We next provide the necessary background for our work by introducing classical planning and decoupled search.

## Classical Planning

Each planning task consists of variables describing states.

**Definition 1** (Variables and States). $\mathcal{V}$ *is a set of* state variables *(primary variables), each* $v \in \mathcal{V}$ *with a finite* domain $D_v$. $\mathcal{D}$ *is a set of binary* derived variables *(secondary variables)* $d \in \mathcal{D}$ *with domain* $D_d = \{0, 1\}$ *and* default value $0$. *A* partial state *is a consistent assignment to variables in* $\mathcal{V} \cup \mathcal{D}$. *A* state *is a complete and consistent assignment to all variables in* $\mathcal{V}$, *and an* extended state *is an assignment to all variables in* $\mathcal{V} \cup \mathcal{D}$. *By* $\mathcal{S}$ *we refer to the set of all states.*

In the context of partial variable assignments, we sometimes denote the variable-value pair $(v, x)$ by writing $v = x$, and for binary variables we also use $v$ to denote the variable-value pair $(v, 1)$ and $\neg v$ to denote $(v, 0)$. For a partial state $p$ we denote the subset of variables defined in $p$ by $V(p) \subseteq \mathcal{V} \cup \mathcal{D}$. Furthermore, we denote $s[L]$ for a partial state $s$ and a set of variables $L \subseteq \mathcal{V} \cup \mathcal{D}$ to represent the restriction/projection of $s$ onto the variables $L$, i.e., $s[L] = \{(v, x) \in s \mid v \in L\}$ and $s(v) = x$ for the assignment of $v$ to $x$ made in $s$.

Planning operators encode the transitions between states.

**Definition 2** (Operators). $\mathcal{O}$ *is a set of* operators *where each operator* $o \in \mathcal{O}$ *is a triplet* $\langle pre(o), \mathit{eff}(o), \mathit{ceff}(o) \rangle$. *The* precondition $pre(o)$ *is a partial state over* $\mathcal{V} \cup \mathcal{D}$, *the* effect $\mathit{eff}(o)$ *is a partial state over* $\mathcal{V}$, *and* $\mathit{ceff}(o)$ *is a set of* conditional effects $(cond \rhd v = x)$, *where* $cond$ *is a partial state over* $\mathcal{V} \cup \mathcal{D}$, $v \in \mathcal{V}$ *is a primary variable, and* $x \in D_v$.[1]

We say that an operator $o \in \mathcal{O}$ *affects* a variable $v \in \mathcal{V}$ if it has an effect on it, formally $v \in V(\mathit{eff}(a))$.

Axioms serve as a means of defining a background theory that describes specific predicates based on other predicates.

**Definition 3** (Axioms). $\mathcal{A}$ *is a set of* axioms $a \in \mathcal{A}$ *of the form* $a = h \leftarrow b$, *where the* head $h$ *is a value assignment of 1 to a derived variable* $d \in \mathcal{D}$, *i.e,* $h = (d, 1)$ *(or just* $h = d$), *and the* body $b$ *is a partial state over primary and secondary variables* $\mathcal{V} \cup \mathcal{D}$.

A set of axioms $\mathcal{A}$ is partitioned into layers $\mathcal{A}_1 \prec \cdots \prec \mathcal{A}_k$. The layer of an axiom is defined by the layer of its head which is determined by a partition of the set of derived variables into subsets $\mathcal{D}_1 \prec \cdots \prec \mathcal{D}_k$. We assume that this partition forms a *stratification*, i.e., that for all $i \in [k]$, and for each $d_i \in \mathcal{D}_i$, it holds that (1) if $d_j \in \mathcal{D}_j$ appears in the body of an axiom with head $d_i$, then $j \leq i$, and (2) if $d_j \in \mathcal{D}_j$ appears with its default value $d_j = 0$ (so its negation $\neg d_j$) in the body of an axiom with head $d_i$, then $j < i$.

The semantics of axioms are defined by the standard stratified semantics (Apt, Blair, and Walker 1988; Thiébaux,

---

[1] We assume well-formed effects, meaning that multiple conditional effects assigning different values to the same variable cannot trigger in the same state, and unconditional effects do not assign different values to variables than the conditional ones.

Hoffmann, and Nebel 2005). Given a state $s \in \mathcal{S}$, the values of the primary variables are preserved, while the values of the derived variables $d \in \mathcal{D}$ are set to their default value (false), i.e., $\neg d$. Then, a fixed-point computation is performed for each axiom layer in turn to determine the final values of the derived variables (Helmert 2009): For each axiom $d \leftarrow b$ in layer $\mathcal{A}_1$, $d$ is set to 1 if $b$ evaluates to true. This process is repeated until no more variable changes occur. The values of the secondary variables defined in layer $i$ are then fixed, and the computation proceeds to the next layer. Finally, the evaluated derived variables together with the state $s$ form the unique extended state $\mathcal{A}(s)$.

With this, we can define a planning task in finite-domain representation as follows (Helmert 2009):

**Definition 4** (FDR Planning Task). *A FDR planning task is a tuple* $\Pi = \langle \mathcal{V}, \mathcal{D}, \mathcal{I}, \mathcal{G}, \mathcal{O}, \mathcal{A} \rangle$, *where* $\mathcal{V}$ *denotes a set of primary variables,* $\mathcal{D}$ *denotes a set of secondary variables,* $\mathcal{I}$ *denotes the* initial state, $\mathcal{G}$ *denotes the partial* goal state, $\mathcal{O}$ *denotes a set of operators, and* $\mathcal{A}$ *denotes a set of axioms.*

An operator $o \in \mathcal{O}$ is *applicable* in a state $s \in \mathcal{S}$ if $pre(o) \subseteq \mathcal{A}(s)$. The result of applying the operator $o$ to a state $s$ is a state $t = s[\![o]\!]$, where $t(v) = x$ for all $(v, x) \in \mathit{eff}(o)$, $t(v) = x$ for all $(cond \rhd v = x) \in \mathit{ceff}(o)$ with $cond \subseteq \mathcal{A}(s)$, and $t(v) = s(v)$ for all variables that do not have such effects. Similarly, we define the application of an operator to partial states $p$. An operator $o$ is applicable in $p$ if $pre(o)[V(p)] \subseteq p$, and the resulting state is defined as $p' = s[\![o]\!][V(p)]$. Based on the semantics of axioms and operators, we can define the state space of a FDR task.

**Definition 5** (FDR State Space). *The* state space *of a FDR planning task* $\Pi$ *is a labeled transition system* $\Theta(\Pi) = \langle \mathcal{S}, \mathcal{O}, T, \mathcal{I}, S_{\mathcal{G}} \rangle$. *The* states $\mathcal{S}$ *are that of* $\Pi$, *and the* transition labels *are the operators* $\mathcal{O}$. *The* initial state *is* $\mathcal{I}$, *and the* goal states *are defined as the set* $S_{\mathcal{G}} = \{s \in \mathcal{S} \mid \mathcal{G} \subseteq \mathcal{A}(s)\}$. *A transition between two extended states* $s \xrightarrow{o} t$ *is contained in* $T$ *if* $o \in \mathcal{O}$, $o$ *is applicable in state* $s$, *and* $t = s[\![o]\!]$.

In this paper, we focus on satisficing planning, aiming to compute any path in the state space of a given FDR planning task from the initial state $\mathcal{I}$ to some goal state $s_{\mathcal{G}} \in S_{\mathcal{G}}$.

A SAS$^+$ planning task is a simplified version of a FDR planning task that does not include derived variables, axioms, or conditional effects (Bäckström and Nebel 1995).

**Definition 6** (SAS$^+$ Planning Task). *A SAS$^+$ planning task is a FDR planning task* $\Pi = \langle \mathcal{V}, \mathcal{D}, \mathcal{I}, \mathcal{G}, \mathcal{O}, \mathcal{A} \rangle$ *where* $\mathcal{D} = \mathcal{A} = \emptyset$ *and for each operator* $o = \langle pre(o), \mathit{eff}(o), \mathit{ceff}(o) \rangle \in \mathcal{O}$, *it holds that* $\mathit{ceff}(o) = \emptyset$.

To simplify the notation, we sometimes denote a SAS$^+$ task as $\Pi = \langle \mathcal{V}, \mathcal{I}, \mathcal{G}, \mathcal{O} \rangle$, and exclude the empty components of a SAS$^+$ operator by representing it as $o = \langle pre(o), \mathit{eff}(o) \rangle$. Moreover, by $V(o) = V(pre(o)) \cup V(\mathit{eff}(o))$, we refer to the variables in the precondition and effect of such operators. It will be convenient to use the concept of the *preimage* $preimg(p', o)$ of a partial state $p'$ and an operator $o$. We define this concept as the set of predecessor partial states $p$ such that they share the same variables $(V(p) = V(p'))$, $o$ is applicable in $p$ $(pre(o)[V(p)] \subseteq p)$, and $p'$ results from the application of $o$ to $p$ $(p' = p[\![o]\!])$.

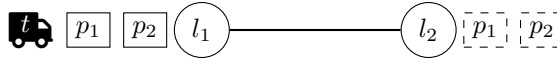

Figure 1: Illustration of the initial state (solid packages) and goal (dashed packages) of the running example.

In the remainder of this paper, we will use the following running example in the form of a SAS$^+$ planning task.

**Example 1** (Running Example). *Let us consider a simple logistics scenario with two connected locations, $l_1$ and $l_2$, along with two packages, $p_1$ and $p_2$, and one truck $t$. These are represented by the variables $\mathcal{V} = \{t, p_1, p_2\}$, with domains: $D_t = \{l_1, l_2\}$ and $D_{p_1} = D_{p_2} = \{l_1, l_2, t\}$.*

*Initially, both the packages and the truck are at position $l_1$, modeled as $\mathcal{I}(v) = l_1$ for all $v \in \mathcal{V}$. The goal is to transport both packages to $l_2$, given as $\mathcal{G} = \{(p_1, l_2), (p_2, l_2)\}$. Both the initial and goal states are illustrated in Figure 1.*

*There are three types of operators in this example:* drive *operators that drive the truck between locations,* load *operators responsible for loading a package onto the truck, and* unload *operators for unloading a package from the truck. Formally, we have for any $i, j \in \{1, 2\}$:*

- drive$(l_i, l_j) = \langle\{(t, l_i)\}, \{(t, l_j)\}\rangle$ *with $j \neq k$*
- load$(p_i, l_j) = \langle\{(t, l_j), (p_i, l_j)\}, \{(p_i, t)\}\rangle$
- unload$(p_i, l_j) = \langle\{(t, l_j), (p_i, t)\}, \{(p_i, l_j)\}\rangle$

*A possible plan is to first load the two packages into the truck, then drive the truck to $l_2$, and unload both packages.*

In Example 1, the number of states grows exponentially as the number of packages grows. This can pose a significant challenge to modern search algorithms.

## Decoupled Search

Decoupled search (Gnad and Hoffmann 2018) is a paradigm for reformulating the state space of SAS$^+$ planning tasks. It can efficiently solve problems like Example 1 by identifying and exploiting causal structure via problem decomposition.

**Definition 7** (Factoring). *Let $\Pi$ be a SAS$^+$ planning task. A pair $\mathcal{F} = \langle C, \mathcal{L}\rangle$ with $\{C\}, \mathcal{L} \subseteq 2^{\mathcal{V}}$ is termed a factoring for $\Pi$ if either $\{C\} \cup \mathcal{L}$ or $\mathcal{L}$ forms a partition of the set of variables $\mathcal{V}$. $C$ represents the (possibly empty) center of $\mathcal{F}$, while $\mathcal{L}$ denotes its leaves.*

Let $\mathcal{F} = \langle C, \mathcal{L}\rangle$ be a factoring for $\Pi$. An operator $o \in \mathcal{O}$ is a *global operator* if there does not exist an $L \in \mathcal{L}$ such that $V(pre(o)) \subseteq C \cup L$ and $V(eff(o)) \subseteq L$. The set of all global operators is denoted $\mathcal{O}^G$. Operators affecting any leaf are called *leaf operators*, denoted $\mathcal{O}^{\mathcal{L}}$.[2] The operators that affect a particular leaf $L \in \mathcal{L}$ are denoted $\mathcal{O}^L$. We define the set of *leaf-only operators* of a leaf $L$ as $\mathcal{O}^L_{\not C} := \mathcal{O}^L \setminus \mathcal{O}^G$. $\mathcal{O}^{\mathcal{L}}_{\not C}$ is the set of all leaf-only operators. A complete assignment to $C$ or to an $L \in \mathcal{L}$ is called a *center state* or *leaf state*, respectively. $S^{\mathcal{L}}$ is the set of all leaf states, and that of a particular leaf $L$ is denoted by $S^L$.

---
[2]An operator can be both a global and a leaf operator.

**Example 2.** *A natural factoring $\mathcal{F}_t$ for the planning task outlined in Example 1 is $\mathcal{F}_t = \langle\{t\}, \{\{p_1\}, \{p_2\}\}\rangle$. Here, the truck forms the center $C = \{t\}$, while each package $p_i$ forms a leaf $L_i = \{p_i\}$. The operators* load *and* unload *are leaf-only operators, with preconditions concerning the truck (center) and the respective package (leaf), and effects concerning the package (leaf) only. Conversely, the truck* drive *operators represent global operators, with preconditions and effects that only affect the truck (center).*

*An alternative factoring, $\mathcal{F}_p = \langle\{p_1, p_2\}, \{\{t\}\}\rangle$, puts the package variables into the center, while assigning the truck to a leaf. Thus, in $\mathcal{F}_p$, the roles of the operators are swapped, i.e., the* drive *operators become leaf-only operators, while the* load *and* unload *operators act as global operators.*

**Definition 8** (Decoupled State). *A decoupled state $s^{\mathcal{D}}$ is a pair $\langle$center$(s^{\mathcal{D}})$, leaves$(s^{\mathcal{D}})\rangle$ where center$(s^{\mathcal{D}})$ is a center state and leaves$(s^{\mathcal{D}}) \subseteq S^{\mathcal{L}}$ is a set of leaf states.*

In essence, a decoupled state $s^{\mathcal{D}}$ represents a collection of explicit states from the original planning task $\Pi$, differing only in the variables of the leaves. A decoupled state $s^{\mathcal{D}}$ satisfies a partial state $p$, denoted by $s^{\mathcal{D}} \models p$, iff (i) $p[C] \subseteq$ center$(s^{\mathcal{D}})$ and (ii) for every $L \in \mathcal{L}$, there exists $s^L \in S^L$ such that $p[L] \subseteq s^L$ and leaves$(s^{\mathcal{D}})(s^L) = 1$.

**Definition 9** (Saturated Decoupled State). *Let $\mathcal{O}^{\mathcal{L}}|_{s^C} := \{o^L \mid o^L \in \mathcal{O}^{\mathcal{L}}_{\not C} \wedge pre(o^L)[C] \subseteq s^C\}$ be the set of of leaf-only operators enabled by a center state $s^C$. For a decoupled state $s^{\mathcal{D}} = \langle s^C, \text{leaves}(s^{\mathcal{D}})\rangle$, $s^{\mathcal{D}}_* = \langle s^C, \text{leaves}^*(s^{\mathcal{D}})\rangle$ is the saturated decoupled state where leaves$^*(s^{\mathcal{D}})$ represents the set of leaf states in the reflexive transitive closure of leaf states reachable from leaves$(s^{\mathcal{D}})$ using $\mathcal{O}^{\mathcal{L}}|_{s^C}$ operators.*

Intuitively, a leaf state $t^L \in$ leaves$^*(s^{\mathcal{D}})$ iff there exists a (possibly empty) sequence of $\mathcal{O}^{\mathcal{L}}|_{s^C}$ operators that transforms a leaf state $s^L \in$ leaves$(s^{\mathcal{D}})$ into $t^L \in S^L$.

With this, we define the decoupled state space as follows.

**Definition 10** (Decoupled State Space). *Let $\Pi$ be a SAS$^+$ planning task and $\mathcal{F} = \langle C, \mathcal{L}\rangle$ a factoring for $\Pi$. The decoupled state space is a labeled transition system $\Theta^{\mathcal{D}}(\Pi, \mathcal{F}) = \langle S^{\mathcal{F}}, \mathcal{O}^G, T^{\mathcal{F}}, \mathcal{I}^{\mathcal{F}}, S^{\mathcal{F}}_{\mathcal{G}}\rangle$ where:*
1. *$S^{\mathcal{F}}$ is the set of all decoupled states.*
2. *The transition labels are the global operators $\mathcal{O}^G$.*
3. *$T^{\mathcal{F}}$ contains a transition $s^{\mathcal{D}} \xrightarrow{o^G} t^{\mathcal{D}} \in T^{\mathcal{F}}$ whenever $o^G \in \mathcal{O}^G$ and $s^{\mathcal{D}}, t^{\mathcal{D}} \in S^{\mathcal{F}}$ such that:*
    (a) *$s^{\mathcal{D}}_* \models pre(o^G)$,*
    (b) *center$(t^{\mathcal{D}}) = $ center$(s^{\mathcal{D}})[\![o^G]\!]$, and*
    (c) *leaves$(t^{\mathcal{D}}) = \{s^L[\![o^G]\!] \mid s^L \in$ leaves$^*(s^{\mathcal{D}}), pre(o^G)[L] \subseteq s^L\}$.*
4. *$\mathcal{I}^{\mathcal{F}} = \langle\mathcal{I}[C], \{\mathcal{I}[L] \mid L \in \mathcal{L}\}\rangle$ is the initial state.*
5. *$S^{\mathcal{F}}_{\mathcal{G}} = \{s^{\mathcal{D}} \in S^{\mathcal{F}} \mid s^{\mathcal{D}}_* \models \mathcal{G}\}$ is the set of goal states.*

**Example 3.** *Consider our running example with the factoring $\mathcal{F} = \mathcal{F}_t$. Part of the decoupled state space is illustrated in Figure 2. In the unsaturated initial state $\mathcal{I}^{\mathcal{F}}$ and its saturated counterpart $\mathcal{I}^{\mathcal{F}}_*$, the single center variable $t$ has the value $l_1$. In $\mathcal{I}^{\mathcal{F}}$, each leaf has a single leaf state, $(p_1, l_1)$ and $(p_2, l_1)$, indicating the initial location $l_1$ of both packages. In the saturated initial state $\mathcal{I}^{\mathcal{F}}_*$,*

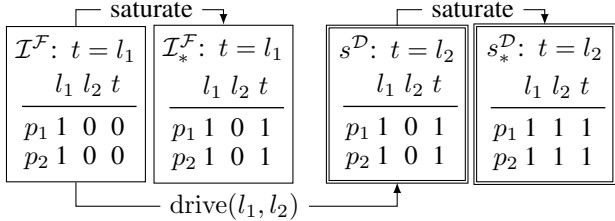

Figure 2: Illustration of the decoupled state space needed to determine a decoupled plan for the running example.

*we have a total of four leaf states, namely* leaves*$(\mathcal{I}^{\mathcal{F}})$ = $\{(p_1, l_1), (p_1, t), (p_2, l_1), (p_2, t)\}$. *This is due to the applicability of leaf-only operators* load$(l_1, p_i)$ *for both packages.*

*Since* $\mathcal{I}_*^{\mathcal{F}} \notin S_{\mathcal{G}}^{\mathcal{F}}$, *we proceed by applying the only applicable global operator,* drive$(l_1, l_2)$, *resulting in the unsaturated decoupled state* $s^{\mathcal{D}}$. *Here, the center variable is updated, while the reached leaf states remain unchanged, since they satisfy the preconditions of the operator and are not affected by it. The saturated decoupled state* $s_*^{\mathcal{D}}$ *matches the goal condition due to the* unload *leaf-only operators, giving us a decoupled plan:* $\langle$drive$(l_1, l_2)\rangle$.

We remark that a decoupled plan – a sequence of labels representing a path from an initial state to a goal state in the decoupled state-space – is not a plan in the original task because it considers only global operators and ignores the leaf-only ones. However, it is efficiently possible to construct a plan for the original task from the decoupled plan by scheduling leaf-only operators along the global ones. In this work we focus on finding decoupled plans, and simply adopt the existing method of plan reconstruction as a post-processing step. For details of this procedure, refer the interested reader to the literature (Gnad and Hoffmann 2018).

## A Novel Approach to Decouple the Search

In this section, we first formalize and exemplify our novel task transformation that generates a FDR task by decoupling a SAS$^+$ planning task based on a given factoring. We prove the correctness of our transformation by showing that the search space of the resulting FDR task is isomorphic to that of decoupled search on the original task. We then derive optimizations to the vanilla encoding that utilize specific causal relationships between factors. Finally, within our new framework, we demonstrate that the approach by Miura and Fukunaga (2017) represents a special form of decoupled search by task transformation. We conclude by showing that our approach generalizes it in multiple dimensions.

### Task Transformation

We define a task transformation called $dec$ that transforms a given SAS$^+$ planning task into a decoupled FDR planning task given a factoring $\mathcal{F}$.

**Definition 11** (Decoupled Transformation). *Let* $\Pi$ = $\langle \mathcal{V}, \mathcal{I}, \mathcal{G}, \mathcal{O} \rangle$ *be SAS$^+$ planning task and* $\mathcal{F}$ = $\langle C, \mathcal{L} \rangle$ *be a factoring for* $\Pi$. *We define the* decoupled transformation *dec as a function that produces a new FDR planning task* $dec(\Pi, \mathcal{F})$ = $\langle \mathcal{V}^{dec}, \mathcal{D}^{dec}, \mathcal{I}^{dec}, \mathcal{G}^{dec}, \mathcal{O}^{dec}, \mathcal{A}^{dec} \rangle$. *The components of this task are detailed below, and together they form the new planning task called* $\Pi_{\mathcal{F}}^{dec}$.

The basic concept behind the task transformation is to embed the leaf state space into the background theory represented by the axioms of the FDR task. The key idea is that an unextended state in $\Pi_{\mathcal{F}}^{dec}$ corresponds to an unsaturated decoupled state for $\Pi$, while an extended state corresponds to a saturated decoupled state. Leaf-only operators, essential for saturating decoupled states, are transformed into axioms.

**Primary variables.** We define a set of primary variables $\mathcal{V}^{dec}$ that consists of the center variables $C$ with their original domain and a binary variable $v_{s^L}$ for every leaf state $s^L \in S^{\mathcal{L}}$. Intuitively, the primary variables describe an unsaturated decoupled state, where the center state is represented by the center variables as in decoupled search and the $v_{s^L}$ variables represent the reached leaf states after the application of a global operator.

$$\mathcal{V}^{dec} = C \cup \{v_{s^L} \mid s^L \in S^{\mathcal{L}}\} \text{ with } D_{v_{s^L}} = \{0, 1\}$$

**Secondary variables.** The secondary variables $\mathcal{D}^{dec}$ consist of three main components. First, we have a derived predicate $d_{s^L}$ for each leaf state. These variables, along with the center primary variables, are used to represent a saturated decoupled state. Second, for each leaf, we have a derived variable to encode whether we have reached a leaf state that satisfies $\mathcal{G}[L]$. Third, similar to the goal condition, for each global operator $o$ and leaf $L$, we have a derived variable used to encode whether we have reached a leaf state $s^L \in S^L$, satisfying the precondition $pre(o)[L] \subseteq s^L$.

$$\mathcal{D}^{dec} = \{d_{s^L} \mid s^L \in S^{\mathcal{L}}\} \cup \{d_{\mathcal{G}[L]} \mid L \in \mathcal{L}, \mathcal{G}[L] \neq \emptyset\} \cup$$
$$\{d_{pre(o)[L]} \mid L \in \mathcal{L}, o \in \mathcal{O}^G, pre(o)[L] \neq \emptyset\}$$

**Initial & Goal states.** Initially, the center variables in $\mathcal{I}^{dec}$ retain the same values as those in $\mathcal{I}[C]$. In addition, for each leaf $L \in \mathcal{L}$, exactly one variable $v_{\mathcal{I}[L]}$ is true, representing the leaf state $\mathcal{I}[L]$. For the partial goal state $\mathcal{G}^{dec}$, the center variables maintain the values present in $\mathcal{G}[C]$. Additionally, for each leaf $L$ with a non-empty goal condition, the goal is expressed by the secondary variable $d_{\mathcal{G}[L]}$ being true.

$$\mathcal{I}^{dec} = \mathcal{I}[C] \cup \{v_{\mathcal{I}[L]} \mid L \in \mathcal{L}\}$$
$$\mathcal{G}^{dec} = \mathcal{G}[C] \cup \{d_{\mathcal{G}[L]} \mid L \in \mathcal{L}, d_{\mathcal{G}[L]} \neq \emptyset\}$$

**Operators.** Each operator $o^{dec} \in \mathcal{O}^{dec}$ in $\Pi_{\mathcal{F}}^{dec}$ corresponds to a global operator $o \in \mathcal{O}^G$ in the original task $\Pi$.

$$\mathcal{O}^{dec} = \{o^{dec} \mid o \in \mathcal{O}^G\}$$

For an operator $o^{dec} \in \mathcal{O}^{dec}$, the preconditions and effects on the center variables remain the same as for $o$. Additionally, we replace the preconditions on the leaf variables in $pre(o)$ with a derived variable $d_{pre(o)[L]}$ for each leaf $L$.

$$pre(o^{dec}) = pre(o)[C] \cup$$
$$\{d_{pre(o)[L]} \mid L \in \mathcal{L}, pre(o)[L] \neq \emptyset\}$$
$$eff(o^{dec}) = eff(o)[C]$$

We use conditional effects to handle the variables that encode reached leaf states. The underlying concept is that these conditional effects copy the reached leaf states from the previous states to the new one, provided that they match the applied global operator. The first set of conditional effects denotes that a decoupled state $t^L$ is reached in the successor state if a state $s^L$ exists such that $t^L = s^L[\![o]\!]$. These effect conditions are on the derived variables $d_{s^L}$, since they correspond to the saturated decoupled state, while the actual effect is on the primary variable $v_{t^L}$, which represents the unsaturated successor state. The second set of conditional effects encode that a leaf state $t^L$ becomes false if there is no leaf state $s^L$ reached such that $t^L = s^L[\![o]\!]$. The latter is necessary because of the closed-world assumption in classical planning, which ensures that primary variables retain their values if they are not affected by an operator.

$$ceff(o^{dec}) = \{(d_{s^L} \rhd v_{t^L}) \mid L \in \mathcal{L}, t^L \in S^L, d_{s^L} \in \mathbf{1}^o_{t^L}\} \cup$$
$$\{(\mathbf{0}^o_{t^L} \rhd \neg v_{t^L}) \mid L \in \mathcal{L}, t^L \in S^L\} \text{ with}$$
$$\mathbf{x}^o_{t^L} = \{(d_{s^L}, \mathbf{x}) \mid s^L \in preimg(t^L, o)\}$$

**Axioms.** The axioms form the final component of $\Pi^{dec}_{\mathcal{F}}$, which consists of four subcomponents for each leaf $L \in \mathcal{L}$.

$$\mathcal{A}^{dec} = \bigcup_{L \in \mathcal{L}} (\mathcal{A}^L_{frame} \cup \mathcal{A}^L_{pre} \cup \mathcal{A}^L_{\mathcal{G}} \cup \mathcal{A}_{\mathcal{O}^L_{\overline{\mathcal{G}}}})$$

After applying a global operator, all derived variable values are reset to false. Thus, the first set of axioms $\mathcal{A}^L_{frame}$ is concerned with restoring the previously reached leaf states, which are stored in the $v_{s^L}$ variables and set accordingly by the conditional effects of the operators $\mathcal{O}^{dec}$. More precisely, the value of $d_{s^L}$ becomes true if $v_{s^L}$ is true.

$$\mathcal{A}^L_{frame} = \{d_{s^L} \leftarrow v_{s^L} \mid s^L \in S^L\}$$

The second and third sets of axioms are used to determine whether a leaf state $s^L$ satisfies the goal condition or the precondition of an operator.

$$\mathcal{A}^L_{\mathcal{G}} = \{d_{\mathcal{G}[L]} \leftarrow d_{s^L} \mid d_{\mathcal{G}[L]} \in \mathcal{D}^{dec}, s^L \in S^L, \mathcal{G}[L] \subseteq s^L\}$$
$$\mathcal{A}^L_{pre} = \{d_{pre(o)[L]} \leftarrow d_{s^L} \mid d_{pre(o)[L]} \in \mathcal{D}^{dec}, s^L \in S^L,$$
$$pre(o)[L] \subseteq s^L\}$$

The fourth and last set of axioms $\mathcal{A}_{\mathcal{O}^L_{\overline{\mathcal{G}}}}$ is concerned with simulating the saturation of a leaf $L$ with leaf-only operators $\mathcal{O}^L_{\overline{\mathcal{G}}}$. We have an axiom for leaf states $s^L, t^L \in S^L$ and leaf-only operator $o \in \mathcal{O}^L_{\overline{\mathcal{G}}}$ if $s^L$ is in the preimage of $t^L$, which implies the applicability of $o$ in $s^L$ such that $t^L = s^L[\![o]\!]$, and if the center preconditions of $o$ are satisfied.

$$\mathcal{A}_{\mathcal{O}^L_{\overline{\mathcal{G}}}} = \{d_{t^L} \leftarrow d_{s^L} \cup pre(o)[C] \mid t^L \in S^L, o \in \mathcal{O}^L_{\overline{\mathcal{G}}}$$
$$s^L \in preimg(t^L, o)\}$$

We next show that $\Pi^{dec}_{\mathcal{F}}$ is a well-formed FDR task (full proof is given in the Appendix).

**Lemma 1.** *Let $\Pi$ be a SAS$^+$ planning task and $\mathcal{F}$ be a factoring for $\Pi$. Then $\Pi^{dec}_{\mathcal{F}}$ is a well-formed FDR planning task.*

| $\mathcal{A}(\mathcal{I}^{dec})$: | | | $\mathcal{A}(s)$: | | |
|---|---|---|---|---|---|
| $\mathcal{I}^{dec}$: $t = l_1$ | | | $s$: $t = l_2$ | | |
| $v_{\{*\}}$ | Val | $d_{\{*\}}$ Val | $v_{\{*\}}$ | Val | $d_{\{*\}}$ Val |
| $(p_1, l_1)$ | 1 | $(p_1, l_1)$ 1 | $(p_1, l_1)$ | 1 | $(p_1, l_1)$ 1 |
| $(p_1, l_2)$ | 0 | $(p_1, l_2)$ 0 | $(p_1, l_2)$ | 0 | $(p_1, l_2)$ 1 |
| $(p_1, t)$ | 0 | $(p_1, t)$ 1 | $(p_1, t)$ | 1 | $(p_1, t)$ 1 |
| $(p_2, l_1)$ | 1 | $(p_2, l_1)$ 1 | $(p_2, l_1)$ | 1 | $(p_2, l_1)$ 1 |
| $(p_2, l_2)$ | 0 | $(p_2, l_2)$ 0 | $(p_2, l_2)$ | 0 | $(p_2, l_2)$ 1 |
| $(p_2, t)$ | 0 | $(p_2, t)$ 1 | $(p_2, t)$ | 1 | $(p_2, t)$ 1 |

extend — drive$(l_1, l_2)$ — extend

Figure 3: Illustration of the state space of the transformed task $\Pi^{dec}_{\mathcal{F}}$ needed to determine a decoupled plan for the running example.

*Proof sketch.* Initial state, goal, preconditions, and unconditional effects are consistent by construction. Further, the conditional effects do not assign conflicting values to the same variable. Finally, a single axiom layer forms a valid stratification, since no secondary variable appears in any axiom body condition with the default value of $0$. □

**Example 4.** *Consider the running example with factoring $\mathcal{F} = \mathcal{F}_t$. Figure 3 illustrates parts of the state space of $\Pi^{dec}_{\mathcal{F}}$. The primary variables include the center variable $t$ and a variable $v_{s^L}$ for each leaf state $s^L$. The secondary variables include a variable $d_{s^L}$ for each leaf state, and variables indicating whether the global operator preconditions or the goal condition are satisfied. The two global operators* drive*, which also form the set $\mathcal{O}^{dec}$, lack preconditions on leaf variables. Thus, there are no secondary variables for the preconditions, there are no $\mathcal{A}^L_{pre}$-axioms, and the conditional effects for both operators simply write the values from the secondary variables $d_{s^L}$ to the primary variables $v_{s^L}$. The goal $\mathcal{G}^{dec} = d_{\{(p_1, l_2)\}} \wedge d_{\{(p_2, l_2)\}}$ refers to the leaf variables $p_1$ and $p_2$. Since we have single variables in the leaves, it follows that $d_{\mathcal{G}[L_i]} = d_{\{(l_i, l_2)\}}$. These variables already exist as they represent leaf states. As a result, the $\mathcal{A}^L_{\mathcal{G}}$-axioms are trivial: $d_{\{(l_i, l_2)\}} \leftarrow d_{\{(l_i, l_2)\}}$. Note that this is not always true, e.g., if both packages were in the same leaf. The transformed task concludes with the frame axioms, which copy values from $v_{s^L}$ to $d_{s^L}$ variables, and the leaf-only operator axioms, representing* load $(d_{(p_i, t)} \leftarrow (p_i, l_j) \wedge (t, l_j))$ *and* unload *operators $(d_{(p_i, l_j)} \leftarrow (p_i, t) \wedge (t, l_j))$.*

*Figure 3 shows the initial state $\mathcal{I}^{dec}$ and its extension, $\mathcal{A}(\mathcal{I}^{dec})$, where the truck and both packages are at $l_1$. In $\mathcal{A}(\mathcal{I}^{dec})$, we can infer that the packages can be at $l_1$ or in the truck. After applying the only applicable operator, we find a goal state $s$ that yields the plan $\langle$drive$(l_1, l_2)\rangle$.*

## Isomorphism of State Spaces

To establish the relationship between the decoupled search space $\Theta^{\mathcal{D}}(\Pi, \mathcal{F})$ and the state space of the transformed planning task $\Theta(\Pi^{dec}_{\mathcal{F}})$, we construct a function that maps between these two transition systems.

**Definition 12** (Mapping Function). *Let $\Pi$ be a SAS$^+$ planning task, $\mathcal{F}$ be a factoring for $\Pi$, $S^{\mathcal{F}}$ be the states of $\Theta^{\mathcal{D}}(\Pi, \mathcal{F})$, and $S^{dec}$ be the states of $\Theta(\Pi^{dec}_{\mathcal{F}})$. We define the function $\varphi : S^{\mathcal{F}} \to S^{dec}$ as $\varphi(s^{\mathcal{D}}) = s$ such that $s[C] = \mathsf{center}(s^{\mathcal{D}})$ and $s(v_{s^L}) = 1$ if $s^L \in \mathsf{leaves}(s^{\mathcal{D}})$ and $s(v_{s^L}) = 0$ otherwise.*

Intuitively, function $\varphi$ establishes a one-to-one correspondence between decoupled states $s^{\mathcal{D}}$ in $\Theta^{\mathcal{D}}(\Pi, \mathcal{F})$ and states $s$ of $\Pi^{dec}_{\mathcal{F}}$. Two states $s^D$ and $s$ are mapped to each other iff they match in the center variables, and a leaf state $s^L$ is reached in $s^{\mathcal{D}}$ iff $v_{s^L}$ is true in $s$. For example, $\varphi(\mathcal{I}^{\mathcal{F}}) = \mathcal{I}^{dec}$ and $\varphi(s^{\mathcal{D}}) = s$ in our running examples as can be seen in Figures 2 and 3.

It is also convenient to establish a relation between a saturated decoupled state and the corresponding extended state of $\Pi^{dec}_{\mathcal{F}}$. Lemma 2 shows that the saturation of a decoupled state $s^{\mathcal{D}}$ and the extension of the corresponding state $\varphi(s^{\mathcal{D}})$ are equivalent. This equivalence is proved by inferring the reachability of the leaf states in $s^{\mathcal{D}}_*$, symbolized by $\mathsf{leaves}^*(s^{\mathcal{D}})$, and in $\mathcal{A}(\varphi(s^{\mathcal{D}}))$, denoted by the variables $d_{s^L}$ (proof is given in the Appendix). Illustrative examples of this equivalence can be observed in Figures 2 and 3, exemplified by $\mathcal{I}^{\mathcal{F}}$ and $\mathcal{I}^{dec}$, as well as $s^{\mathcal{D}}$ and $s$.

**Lemma 2.** *Let $s^{\mathcal{D}} \in S^{\mathcal{F}}$ be a decoupled state and $s^L \in S^{\mathcal{L}}$ a leaf state. Then $s^L \in \mathsf{leaves}^*(s^{\mathcal{D}})$ iff $\mathcal{A}(\varphi(s^{\mathcal{D}}))(d_{s^L}) = 1$.*

Finally, we show that the decoupled search space and the state space of the transformed planning task are isomorphic. This implies that search algorithms applied to the transformed planning task will behave identically to their specialized counterparts designed for decoupled search. For the detailed proof, we refer the reader to the Appendix.

**Theorem 1.** *Let $\Pi = \langle \mathcal{V}, \mathcal{I}, \mathcal{G}, \mathcal{O} \rangle$ be a SAS$^+$ planning task and $\mathcal{F}$ be a factoring for $\Pi$. Then the FDR state space of $\Pi^{dec}_{\mathcal{F}}$ and the decoupled state space of $\Pi$ are isomorphic, i.e., $\Theta(\Pi^{dec}_{\mathcal{F}}) \sim \Theta^{\mathcal{D}}(\Pi, \mathcal{F})$.*

*Proof sketch.* Let $\Theta^{\mathcal{D}}(\Pi, \mathcal{F}) = \langle S^{\mathcal{F}}, \mathcal{O}^G, T^{\mathcal{F}}, \mathcal{I}^{\mathcal{F}}, S^{\mathcal{F}}_{\mathcal{G}} \rangle$ and $\Theta(\Pi^{dec}_{\mathcal{F}}) = \langle S^{dec}, \mathcal{O}^{dec}, T^{dec}, \mathcal{I}^{dec}, S^{dec}_{\mathcal{G}} \rangle$. Function $\varphi$ is bijective since it establishes a one-to-one mapping between the different state sets. Furthermore, it holds that 1. $\varphi(\mathcal{I}^{\mathcal{F}}) = \mathcal{I}^{dec}$, 2. $s^{\mathcal{D}} \in S^{\mathcal{F}}_{\mathcal{G}}$ iff $\varphi(s^{\mathcal{D}}) \in S^{dec}_{\mathcal{G}}$, and 3. $s^{\mathcal{D}} \xrightarrow{o} t^{\mathcal{D}} \in T^{\mathcal{F}}$ iff $\varphi(s^{\mathcal{D}}) \xrightarrow{o^{dec}} \varphi(t^{\mathcal{D}}) \in T^{dec}$. □

## Optimizations: Operator and Leaf Types

In the introduced task transformation, every global operator and every leaf is treated in the same way, regardless of the underlying structure. This results in an encoding that can contain many conditional effects to represent the semantics of the reached leaf states. However, we can exploit the fact that certain global operators have varying and sometimes no influence on the reachability of leaf states for a particular leaf. This insight will help to derive a more compact encoding of the effects of global operators on leaves.

**Definition 13** (Irrelevant Operator). *For a SAS$^+$ task $\Pi$ and a factoring $\mathcal{F} = \langle C, \mathcal{L} \rangle$, a global operator $o^G \in \mathcal{O}^G$ is $L$-irrelevant for a leaf $L \in \mathcal{L}$ iff both of the following conditions hold: (1) $V(o^G) \cap L = \emptyset$, and (2) $V(o^G) \cap V(pre(o^L)) = \emptyset$ for all leaf-only operators $o^L \in \mathcal{O}^L_{\emptyset}$.*

Intuitively, applying an $L$-irrelevant global operator $o^G$ does not affect the reachability within a leaf $L$ in any way. Consequently, there's no need to transfer the inferred values from $d_{s^L}$ to $v_{s^L}$ for the successor state, since these values can be inferred again from the unchanged $v_{s^L}$ variables. This shows that we can omit conditional effects on leaves $L$ for which an operator is considered $L$-irrelevant.

Next, we introduce the concept of conclusive operators and conclusive leaves.

**Definition 14** (Conclusive Operator). *For a SAS$^+$ planning task $\Pi$ and a factoring $\mathcal{F} = \langle C, \mathcal{L} \rangle$, a global operator $o^G \in \mathcal{O}^G$ is $L$-conclusive for a leaf $L \in \mathcal{L}$ iff $V(o^G) \cap L = L$.*

**Definition 15** (Conclusive Leaf). *For a SAS$^+$ planning task $\Pi$ and a factoring $\mathcal{F} = \langle C, \mathcal{L} \rangle$, a leaf $L \in \mathcal{L}$ is conclusive iff it holds that each global operator $o^G \in \mathcal{O}^G$ is either $L$-conclusive or $L$-irrelevant.*

The idea behind Definition 14 is that after applying an $L$-conclusive global operator, exactly one leaf state $s^L \in S^L$ is reached, since all variables $L$ are conclusively fixed by the operator effect or precondition. With that, for an $L$-conclusive leaf (Definition 15), the application of a global operator will either uniquely fix all variable values of $L$ or not affect the reachability within $L$. As initially only a single leaf state is reached in each leaf, this eliminates the need for a separate variable $v_{s^L}$ for each leaf state within a conclusive leaf $L$. A single leaf state is sufficient to infer the reachable leaf states of $L$ at any point, hence we can represent this state in a factored way using the original primary variables of $L$ and adapt $\mathcal{A}^L_{frame}$ to refer to these in the body.

Finally, for global operators $o^G \in \mathcal{O}^G$ that have no preconditions and effects on a leaf $L$, along any transition $s^{\mathcal{D}} \xrightarrow{o^G} t^{\mathcal{D}}$ if $s^L \in \mathsf{leaves}(s^{\mathcal{D}})$ then $s^L \in \mathsf{leaves}(t^{\mathcal{D}})$. We can exploit this by dropping the conditional effects of $o^G$ that make any variable $v_{s^L}$ false.

**Example 5.** *Consider our running example with $\mathcal{F}_t$. Here, the* drive *operators are neither conclusive nor irrelevant to the two leaves. The leaf variables are not mentioned in the operators, but the position of the truck $t$, which is affected by these operators, appears in the preconditions of the leaf-only operators. However, both leaves $\{p_1\}$ and $\{p_2\}$ are fork leaves, so the global* drive *operators have no precondition or effect on the leaves. Thus, we can omit the conditional effects for those operators that make the $v_{s^L}$ variables false.*

*Now let us reconsider the example with an additional truck and a factoring similar to $\mathcal{F}_p$, where the two packages are the center, and the two trucks form individual leaves. When a global* load *operator is applied to load a package onto truck $t_1$, its influence is only on the reachability of leaf $\{t_1\}$, by requiring that the truck be positioned at the same location as the package. So this operator is $\{t_1\}$-conclusive and $\{t_2\}$-irrelevant. Thus, for this particular operator, there is no need to encode the conditional effect associated with the $\{t_2\}$ leaf. Finally, all leaves are actually conclusive, since all* load *and* unload *operators are either irrelevant or conclusive for each leaf.*

**Miura and Fukunaga (2017)** describe methods to transform a given planning task into a more concise representation by introducing derived predicates and axioms. They propose two methods, one based on mutexes, which we will not discuss further as it is orthogonal to our work, and a method called $\tau$-axiom extraction that identifies operators with specific properties. This method turns these operators into axioms and casts their effect variables into derived variables. More precisely, their approach searches for a cardinality maximal set of variables $L \subseteq \mathcal{V}$, where each operator $o$ either affects only variables of $L$, i.e., $V(\mathit{eff}(o)) \subseteq L$, and is then transformed into axioms, or determines all values of $L$, i.e., $L \subseteq V(\mathit{pre}(o))$, and remains an operator. A closer look reveals that this is equivalent to searching for a set of global operators that are $L$-conclusive for a single leaf.

Overall, this means that the approach introduced by Miura and Fukunaga (2017) is a special case of our decoupled task transformation. The approach presented in this paper extends their concept in several dimensions, embodying decoupled search in its full generality: allowing arbitrary and multiple leaves instead of a single conclusive one, and allowing arbitrary global operators instead of supporting exclusively conclusive ones.

## Experiments

We implemented our decoupled task transformation in the Fast Downward 23.06 framework (FD) (Helmert 2006). Our experiments were conducted on a cluster of Intel Xeon Gold 6130 CPUs using Downward Lab 8.0 (Seipp et al. 2017), with runtime and memory limits of 30 min and 8 GiB, on all 2106 STRIPS instances from the satisficing sequential tracks of the International Planning Competitions 1998–2023. Our code and experimental data are available online.[3]

We extended FD's task transformation interface for our own transformation, such that we can (1) run a search directly on the transformed task, or (2) write the transformed task to disk in (grounded) PDDL or FD's own `*.sas` format. To reconstruct full plans from the obtained global-operator sequences, we integrated the solution reconstruction of the decoupled-search planner of Gnad and Hoffmann (2018). In the following, we evaluate our approach by performing search directly on the transformed task with two different configurations: lazy greedy best-first search (GBFS) with the $h^{\mathrm{FF}}$ heuristic (Hoffmann and Nebel 2001) and a dual-queue open list with preferred operators (PO) (Richter and Helmert 2009), and the first iteration of LAMA (Richter and Westphal 2010). We always use an operator cost of one and impose the 30-min runtime limit on the entire process, i.e., transformation *and* search.

We compare our decoupled task representation with the outlined optimizations ($dec$) to the original SAS$^+$ encoding of FD ($sas$). To see the effect of the optimizations, we also show data for the non-optimized basic transformation ($dec^\varnothing$). Furthermore, we compare to the native decoupled-search implementation of Gnad and Hoffmann (2018) ($gh$), and the transformation by Miura and Fukunaga (2017) ($mf$). For the latter we reimplemented their variable-based axiom

---

[3]Link removed to ensure anonymous submission.

| Time | <1s | <5s | <10s | <30s | <60s | ≥60s | DNF |
|---|---|---|---|---|---|---|---|
| # Inst. | 957 | 48 | 22 | 13 | 4 | 11 | 4 |

Table 1: Decoupled transformation runtime statistics.

extraction in our planner, which serves as a factoring for the transformation. Finally, we include the Merge-And-Shrink task reformulation method proposed by Torralba and Sievers (2019) ($ts$), which to our knowledge is the only alternative technique that extensively restructures the state space.

As factoring strategy for our transformation and native decoupled search, we pick the best configuration reported by Gnad, Torralba, and Fišer (2022) for satisficing planning, called F20s, giving it a time limit of 30s. As a minor modification, we restrict the set of potential leaf factors to variable sets with domain-size product smaller than one million. The original strategy uses $2^{32}$ as limit, but we observed that too large leaf factors incur a significant overhead in our transformation. We say that the strategy is *successful* if it terminates in the limits and results in a factoring with at least two leaf factors. Otherwise, like prior work on decoupled search, we *abstain* from solving the task, assuming that a linear search-space reduction does not usually pay off. We use the IBM CPLEX solver in version 12.10 to compute the factorings.[4]

**Transformation statistics.** Table 1 shows runtime statistics of our transformation. When the factoring is successful, the transformation takes negligible time in most cases, finishing in less than 10 seconds for 97% of the instances. The maximum runtime is 600 seconds. There are only 4 instances in which the transformation runs out of time or memory. In Figure 4 we analyze the task sizes under the transformation. The size of a task is measured as the encoding size in the same way as this is done by FD's translator component (Helmert 2009). The left plot compares the original encoding to the optimized decoupled task. As expected, the transformation can lead to a significant increase in the encoding size, up to more than four orders of magnitude. The majority of instances only sees a moderate increase of up to a factor of 10, though. The right plot shows that our optimizations are indeed effective in reducing the encoding size, yielding savings of almost four orders of magnitude. We highlight the ratio of conclusive leaves over the total number of leaves in different colors/shapes. This nicely illustrates that if most leaves are conclusive then the transformed task is usually only larger by a constant factor than the original task. That is because our optimizations effectively reduce the encoding size in that case, as seen in the right plot.

**Planning performance.** Considering the factoring that embodies the $mf$ approach, we can observe that it is not applicable in the vast majority of instances due to the restriction to a single conclusive leaf. On our benchmark set, the $mf$ approach is effective on 311 instances, of which 271 are solved with GBFS and $h^{\mathrm{FF}}$, respectively 307 with LAMA. The $sas$ baseline solves 269, respectively 307 instances as well on that instance set, so performs very similarly in terms

---

[4]https://www.ibm.com/analytics/cplex-optimizer

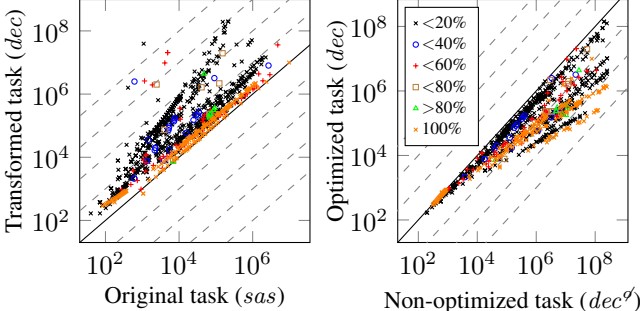

Figure 4: The plots compare task sizes on a per-instance basis. The left plot compares the original task size to that of the optimized transformed task, the right plots compares the non-optimized to the optimized encoding. The different colors indicates the ratio of leaves that are conclusive.

of coverage. Runtime-wise, we see a maximum speed-up factor of 337; in terms of arithmetic/geometric mean $mf$ is faster than $sas$ by a factor of 2.18/1.03. However, it turns out that some strategies reported in Gnad, Torralba, and Fišer (2022) result in factorings that outperform $mf$ when a single leaf factor is considered, leading to a larger reduction.

Turning to the full benchmark set, Table 2 shows coverage results (number of instances solved) for all instances where F20s is successful (omitting the already discussed results for $mf$). For GBFS with $h^{FF}$ and PO we observe a huge impact of our encoding optimizations, with +75 solved instances distributed over many domains. While our transformation-based approach is generally behind the native implementation ($gh$), it clearly outperforms the original encoding ($sas$), even if that uses LAMA. Notably, our approach beats $gh$ in two domains. Compared to the Merge-And-Shrink reformulation ($ts$), either of the methods outperforms the other in some domains, in total $dec$ is ahead by 29 instances. We remark that both $gh$ and $ts$ would require a specialized adaptation of the landmark heuristic in LAMA, whereas our approach works out of the box. When using LAMA, our approach shows particularly good results in childsnack, floortile, and nomystery, and beats the baseline with SAS$^+$ encoding by 22 instances overall.

## Conclusion

We introduced a novel task transformations for classical planning that exactly mimics the behavior of decoupled state space search, emphasizing the power of task reformulations. Our transformation works for arbitrary tasks in SAS$^+$ format, encoding the leaf dynamics of decoupled search as axioms into a FDR task. We prove the correctness of our transformation by showing that the state space of the transformed task is isomorphic to the decoupled state space when employing the same problem decomposition. This allows any search technique to be applied to the decoupled task without specific adaptation, opening up numerous possibilities for the use of decoupled search. In our evaluation, we demonstrate this using the well-known LAMA planner.

While the overall search performance on the transformed

| Domain | | GBFS($h^{FF}$, PO) | | | | | LAMA | |
|---|---|---|---|---|---|---|---|---|
| | | $dec^{g}$ | $dec$ | $sas$ | $gh$ | $ts$ | $dec$ | $sas$ |
| airport | 25 | 9 | 12 | **14** | 11 | 13 | **12** | 11 |
| childsnack | 20 | **20** | **20** | 7 | **20** | 8 | **20** | 6 |
| data-network | 20 | 6 | 9 | 10 | 5 | **11** | 10 | **13** |
| depot | 22 | 20 | 21 | 18 | 21 | **22** | 21 | 20 |
| elevators-11 | 20 | **20** | **20** | 19 | **20** | **20** | **20** | **20** |
| floortile-11 | 20 | 13 | 14 | 8 | **17** | 8 | **19** | 7 |
| floortile-14 | 20 | 14 | 14 | 2 | **20** | 5 | **20** | 2 |
| grid | 5 | **5** | **5** | 4 | **5** | **5** | **5** | **5** |
| hiking | 18 | **18** | **18** | **18** | **18** | 17 | **18** | **18** |
| logistics98 | 35 | 30 | 32 | 33 | **35** | **35** | 31 | **35** |
| maintenance | 4 | **1** | **1** | **1** | 0 | 0 | 0 | **1** |
| mystery | 7 | **4** | **4** | **4** | 3 | **4** | **4** | **4** |
| nomystery | 20 | 15 | 16 | 9 | **19** | 10 | **18** | 12 |
| openstacks-11 | 20 | 11 | 18 | **20** | **20** | **20** | **20** | **20** |
| openstacks-14 | 20 | 0 | 14 | **20** | **20** | 16 | **20** | **20** |
| organic-split | 15 | 6 | 8 | **11** | 10 | 3 | 10 | **14** |
| pathways | 30 | **23** | **23** | 21 | **23** | **23** | 22 | **23** |
| quantum-layout | 20 | 18 | 19 | 19 | 19 | **20** | **20** | **20** |
| recharging-robots | 15 | 6 | 12 | 11 | **14** | 13 | 12 | **13** |
| rovers | 40 | 36 | 39 | **40** | **40** | **40** | 39 | **40** |
| satellite | 36 | **36** | **36** | **36** | **36** | 31 | **36** | **36** |
| scanalyzer-08 | 3 | 0 | 0 | **3** | **3** | **3** | 0 | **3** |
| scanalyzer-11 | 3 | 0 | 0 | **3** | **3** | **3** | 0 | **3** |
| slitherlink | 3 | 0 | 0 | 1 | **2** | 0 | 0 | **1** |
| tetris | 17 | 1 | 11 | **14** | 11 | 2 | 8 | **14** |
| tidybot-11 | 18 | 14 | 14 | **16** | 14 | 15 | 15 | **16** |
| transport-08 | 30 | **30** | **30** | 28 | **30** | **30** | **30** | **30** |
| transport-11 | 20 | 13 | **20** | 11 | **20** | **20** | **20** | 19 |
| transport-14 | 20 | 6 | **20** | 9 | **20** | **20** | **20** | 17 |
| trucks | 30 | 13 | 13 | **19** | 18 | 16 | 14 | **17** |
| woodwork-11 | 20 | 19 | 19 | **20** | **20** | **20** | 18 | **20** |
| others | 463 | 463 | 463 | 463 | 463 | 463 | 463 | 463 |
| **Sum** | 1059 | 870 | 945 | 912 | **980** | 916 | **965** | 943 |

Table 2: Coverage of GBFS with $h^{FF}$ and preferred operators, respectively LAMA, projected on the set of instances in which our factoring method is successful. Best coverage is highlighted in bold face.

task may fall slightly behind a native decoupled search implementation, we observe that it is competitive on many domains. Depending on the properties of the factoring, it occasionally even outperforms the native approach. As a result, planners of different kinds can now be "automatically decoupled" while maintaining near-native performance.

For future work, we plan to further investigate approaches that reduce the size of the transformed task. This could be achieved by employing the irrelevance pruning of Torralba et al. (2016), which admissibly prunes the leaf state spaces. Another interesting question is whether it is feasible to adapt our approach to be suitable for optimal planning. If leaf operators have no costs, our current transformation ensures optimality. However, in the general scenario, tracking leaf-operator costs along with derived predicates remains an open question. Finally, we are curious to see if it is possible to encode other reduction techniques like symmetry breaking (Domshlak, Katz, and Shleyfman 2012) or partial-order reduction (Wehrle and Helmert 2012) as a task transformation.

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
