# OpenReview forum: "Decoupled Search for the Masses: A Novel Task Transformation for Classical Planning"
_icaps-conference.org/ICAPS/2024/Conference — ICAPS 2024_

### Official Review · Reviewer_v6d9 · 2024-01-22

**Significance And Importance:** 2
**Soundness:** 4
**Novelty:** 3
**Clarity:** 4
**Overall Evaluation:** 3
**Confidence:** 3

**Weaknesses:**

1: Minor weaknesses that are easily fixable.

**Contributions Of The Paper:**

The paper provides a translation of decoupled search for SAS+ planning problems into FDR state space search, suitable for consumption by major planners, such as Fast Downward. The paper also provides optimizations which make the translation competitive with direct implementations of decoupled search. The existence of translations such as this are important for future research, as its provides a way for experimenting with decoupled search in extensions to classical planning, such as numeric (at least if the numeric variables were kept to the center), and allows for experimenting with combinations of decoupled search with heuristics and other search techniques that can be difficult and time consuming to reimplement in native decoupled-search planners.

**Ethical Considerations:**

(1) Not Applicable: The paper does not have any ethical considerations to address

**Nomination For Best Paper:**

Yes

**Questions For Authors:**

1. The gain against LAMA and GBFS was highly concentrated in a few domains. Are there clear metrics from the translation which explain those results? That would be a benefit in portfolio approaches to planning.

Post-rebuttal:
Thank you for the discussion

**Reproducibility:**

4: Authors promise to release code and domains (whichever apply).

**Strengths Of The Paper:**

Clarity:
Overall, the text was well-organized. The definitions and theorems were crisp and clear. For the denser mathematical constructs, the paper provided examples to work through the use and implications of the definitions and theorems. Presenting complex technical work like this is difficult, and this paper serves as an exemplar in communicating highly-mathematical advances.

Comparison with related work:
The theoretical and experimental comparison against the work of Miura and Fukunaga was an especially edifying read, as their work had not been previously related to decoupled search.

**Weaknesses Of The Paper:**

Evaluation (minor):
The optimizations were significantly important in some domains, such as transport and openstacks. Understanding why that's the case would help guide future work, such as if there's the possibility to guide the decoupled representation construction in a way that prefers conclusive leaves.

Errata:
L210: j \neq k => i \neq j
L261: There's a use of the leaves(...)(...) syntax that isn't explained or defined, but matches the prices syntax in recent decoupled search papers.

---

> ### Author Rebuttal · Authors · 2024-01-26
>
> Thank you for your comments and questions.
>
> > 1. The gain against LAMA and GBFS was highly concentrated in a few domains. Are there clear metrics from the translation which explain those results? That would be a benefit in portfolio approaches to planning.
>
> In most domains where the gains were high, the task was highly "decouplable", meaning that we found factorings with many leaves. One such example is transport, where each package or truck corresponds to a separate leaf. On instances with high decouplability, the main factors that influences the search performance in the transformed task are related to the leaf structure, i.e., compatibility, leaf-irrelevance, and the other described optimization. These properties should quite reliably predict the performance. (If you are interested, please also see our response to reviewer 16JE, which goes in a similar direction.)

---

### Official Review · Reviewer_nApQ · 2024-01-22

**Significance And Importance:** 3
**Soundness:** 4
**Novelty:** 2
**Clarity:** 3
**Overall Evaluation:** 2
**Confidence:** 3

**Weaknesses:**

2: No major or minor weaknesses.

**Contributions Of The Paper:**

This research investigates the impact of problem representation on classical planning tasks, emphasizing the significance of problem reformulation for enhanced performance. While domain-specific scenarios often benefit from targeted reformulations, this study focuses on domain-independent planning. The merge-and-shrink task reformulation, designed for factored transition systems, exemplifies altering state space to facilitate search. The paper introduces decoupled state-space search as a task reformulation. Unlike existing techniques, decoupled search is computed for non-optimal planning within the FDR. The study demonstrates the isomorphism between FDR planning tasks and decoupled search, allowing the application of diverse planning technology. Experimental results reveal the competitiveness and versatility of this approach across a comprehensive set of IPC benchmarks, showcasing its competiveness compared to alternative approaches and other reformulation techniques.

**Ethical Considerations:**

(1) Not Applicable: The paper does not have any ethical considerations to address

**Nomination For Best Paper:**

No

**Questions For Authors:**

I have no particular question for the authors. I think this is a particularly well-conducted piece of work, well positioned in relation to the bibliography, even if the results are mixed.

1 - Could you please make explicit the meaning of 1^{O}_{t^L} and 0^{O}_{t^L} (line 390)?

**Reproducibility:**

4: Authors promise to release code and domains (whichever apply).

**Strengths Of The Paper:**

- Experiments are really exhaustive and convincing.

**Weaknesses Of The Paper:**

- Notations are plethoric and really hard to follow (for instance def. 9). In my opinion, many notations fall into over-formalization with indices, exponents, brackets, square brackets, parentheses, starts, \cal letters, arrows etc. Fortunately, the examples are very helpful.

---

> ### Author Rebuttal · Authors · 2024-01-26
>
> Thank you for your comments and questions.
>
> > 1. Could you please make explicit the meaning of 1^{O}_{t^L} and 0^{O}_{t^L} (line 390)?
>
> Thank you for pointing out that this definition/concept was difficult to follow. We will make it more explicit in the final version of the paper. The sets $1_{t^L}^{o}$ and $0_{t^L}^{o}$ describe partial states over derived variables (variable-value pairs). The derived variables contained in the set correspond to all predecessor leaf states $s^L$ that have a transition with the operator $o$ to the leaf state $t^L$ for a leaf $L$. Finally, $1_{t^L}^{o}$ assigns the true value 1 to these derived variables, and $0_{t^L}^{o}$ assigns the false value 0 to them.

---

### Official Review · Reviewer_16JE · 2024-01-23

**Significance And Importance:** 3
**Soundness:** 4
**Novelty:** 3
**Clarity:** 4
**Overall Evaluation:** 2
**Confidence:** 3

**Weaknesses:**

2: No major or minor weaknesses.

**Contributions Of The Paper:**

The paper shows that decoupled state space search can be interpreted as a task reformulation in classical planning, and they show this reformulation in terms of a translation from the SAS+ problem to a decoupled FDR planning problem. The decoupled search translation is of course problem independent.

The authors show that decoupled search for suboptimal planning can be obtained by a task transformation within the widely used finite-domain representation (FDR) formalism. This generalises previous work by extending the applicability of decoupled search to a wider range of planning technologies.

The approach presented in the paper also generalises a previous task reformulation technique by Miura and Fukunaga. The authors demonstrate that their approach not only encompasses the previous technique, but also offers additional advantages and competitiveness over other related problem reformulation techniques.

**Ethical Considerations:**

(1) Not Applicable: The paper does not have any ethical considerations to address

**Nomination For Best Paper:**

Yes

**Questions For Authors:**

1) Work has been done in the area of extracting macros from the inherent structure of the planning problem. Can the decoupled task reformulation be facilitated by an initial transformation of the problem, associating actions into macros? I have in mind in particular the aggregation, if necessary, of leaf-only operators and/or global operators.

2) Certain benchmarks "resist" to the approach, requiring a longer runtime. Have you analysed the reasons of that, and can you provide a way to classify planning instances in order to evaluate their degree of "decouplability"?

**Reproducibility:**

4: Authors promise to release code and domains (whichever apply).

**Strengths Of The Paper:**

This paper seems flawless to me, perhaps because of my limited in-depth knowledge of decoupled search approaches. I've checked the proofs and examples and can't find any technical errors. In my opinion, the paper is well written, and thanks to the examples provided, it is clear enough.

The approach is interesting for Planning applications, modulo the plan reconstruction methods that need to be applied in order to rebuild he plan for the original task from the decoupled plan.

**Weaknesses Of The Paper:**

These are minor remarks about the paper, please consider them for the eventual camera ready:

l.210  with j \neq i
Def. 7: Strictly speaking, a partition cannot be empty.

---

> ### Author Rebuttal · Authors · 2024-01-26
>
> Thank you for your comments and questions.
>
>
> > 1. Work has been done in the area of extracting macros from the inherent structure of the planning problem. Can the decoupled task reformulation be facilitated by an initial transformation of the problem, associating actions into macros? I have in mind in particular the aggregation, if necessary, of leaf-only operators and/or global operators.
>
> This certainly sounds interesting and, in our opinion, is a promising avenue for future work. We think it is possible to generate such macros when avoiding aggregation of actions from different leaves. Thus, aggregating leaf actions from one leaf should be readily possible. It may also be possible to consider a combination of global actions and actions from a specific leaf. Such macros should not affect the factoring, but can potentially reduce the size of the transformed task.
>
>
> > 2. Certain benchmarks "resist" to the approach, requiring a longer runtime. Have you analysed the reasons of that, and can you provide a way to classify planning instances in order to evaluate their degree of "decouplability"?
>
> In an ideal setting, the gains of decoupled search scale exponentially with the number of leaves. Therefore, most methods from the literature for finding a good factoring aim at maximizing the number of leaves. In most domains where decoupled search in any version, native or via task transformation, performs worse than search on the undecoupled SAS task, this is due to a small number of leaves (in most cases only two). In this case, the per-node overhead of decoupled search does not always pay off. One can quite reliably predict good performance based on properties such as the number of leaves or, more relevant for the transformation, if those leaves have special properties such as being conclusive. (If you are interested, please also see our response to reviewer v6d9, which goes in a similar direction.)

---

### Meta-Review · Area_Chair_wyaW · 2024-02-05

**Recommendation:** Accept (Oral)
**Confidence:** 4

**Metareview:**

Keeping this short: this is a clear accept. All reviewers liked the work, and we look forward to seeing it as ICAPS. Because everyone was in (positive) agreement, the paper did not attract further discussion, so I have nothing to add to the reviews. Congratulations on a very nice paper!

**Ethical Considerations:**

(1) Not Applicable: The paper does not have any ethical considerations to address